# High-Temperature Aroma Mitigation and Fragrance Analysis of Ethyl Cellulose/Silica Hybrid Microcapsules for Scented Fabrics

**Zuobing Xiao** [1,2]**, Bin Zhang** [1]**, Xingran Kou** [1]**, Yunwei Niu** [1]**, Liu Hong** [3]**, Wei Zhao** [3]**, Haocheng Cai** [3] **and Xinyu Lu** [1,*]

1   Department of Perfume and Aroma Technology, Shanghai Institute of Technology, Shanghai 201418, China; xzb@sit.edu.cn (Z.X.); zzb97zzb@163.com (B.Z.); kouxr@sit.edu.cn (X.K.); nyw@sit.edu.cn (Y.N.)
2   School of Agriculture and Biology, Shanghai Jiao Tong University, Shanghai 200240, China
3   China Tobacco Yunnan Industrial Co., Ltd., Kunming 650231, China; 13122028951@163.com (L.H.); zhaoweizhaorong@126.com (W.Z.); chc19901026@163.com (H.C.)
*   Correspondence: luxinyu@sit.edu.cn

**Abstract:** Microencapsulation can improve the thermal stability of a fragrance, and composite wall materials are one way to further improve the thermal stability of microcapsules. This paper presents a facile approach for cotton fabric coatings by using cellulose/silica hybrid microcapsules. Lavender fragrance oil-loaded cellulose/silica hybrid microcapsules were one-step synthesized via emulsion solvent diffusion. The prepared microcapsules were found to be spherical in shape with a particle size distribution between 500 to 1000 nm. Due to the slow releasing of lavender fragrance oil in the capsules, the fragrance loss rate of (3-aminopropyl)triethoxysilane (APTES)-, triethoxy(3-glycidyloxypropyl)silane (GPTES)-, and (3-aercaptopropyl)trie-thoxysilane (MPTES)-modified cellulose/silica hybrid microcapsules are 25.2%, 35.1%, and 16.7% after six hours at 120 °C. E-nose and gas chromatography–mass spectrometry (GCMS) studies found that the fragranced cotton fabrics had good retention of characteristic aromas. It provides the basis for the application of the heating treatment of cotton fabrics in sterilization, bleaching, printing, and other processes.

**Keywords:** microcapsules; heat resistant; composite material; fragrance



## 1. Introduction

As an increasingly indispensable substance in daily life, fragrance can make people feel happy and improve their quality of life and work efficiency [1,2]. Fragrance components are extremely volatile with a short retention time [3] and weak heat resistance. Therefore, microencapsulation of aromas is a better option [4–7]. However, in the application of hot processing, the aroma retention rate of microcapsules of conventional materials at a high temperature is still low. In the past, people have never given up the preparation of microcapsules with good thermal stability [8–10].

Compared with single-layer microcapsules, double/multi-layer microcapsules and composite-layered microcapsules have relatively better thermal stability, and double-layered microcapsules can compensate for the loss of fragrance caused by the pores in the shell material [11–13]. Composite shell materials are more often in the form of organic and inorganic nanoparticles; special chemical bonds are formed between nanoparticles and organic materials, and the combination between particles and macromolecules reduces the number and freedom of macromolecular side chains, enhancing their stability and reducing fragrance loss [14]. Moreover, a key factor affecting the heat resistance of composites is the interaction between different nanomaterials and polymers [15,16]. On the other hand, inorganic nanoparticle–organic composites have the advantages of both inorganic and organic shell materials, making the preparation process of microcapsules relatively simple, non-hazardous, and more widely used, and the technology is more mature and can cover

more kinds of core materials, such as drugs, fragrances, dyes, catalysts, phase-change materials, etc [17–20]. Cotton fabrics with different filler decoration have multiple applications. The addition of different coatings to the fabric reduces the "void-filler" inside the fabric and enhances the conductive, antibacterial, and ultraviolet shielding properties of the fabric; for example, as the fabric is impregnated in the electromagnetic interference shielding coating for longer periods, the shielding properties can be increased to their peak [21,22].

Ethyl cellulose (EC) is one of the most widely used water-insoluble cellulose, which is easily degradable, low cost, strong alkali and weak acid resistant, and has good thermal stability. In addition, the decomposition of EC at a high temperature, without an unpleasant odor, and the high toughness of EC films at high temperature are ideal materials for the preparation of microcapsules with excellent thermal stability [23–26]. Nano-silica (nano-$SiO_2$) microcapsules have the advantages of excellent biocompatibility and a wide range of applications, and their composite modification can produce synergistic effects and endow the composites with new properties [27–30]. The large number of hydroxyl groups on the EC backbone lends itself to chemical modification by oxidation, silylation, and electrostatic interaction reactions. Placing epoxy groups, amine groups, -SH, and hydroxyl groups at the ends and sides of the copolymer is essential for high temperature applications [31]. Modifying the surface of silica particles with -$NH_2$ and -SH groups is beneficial to the adsorption of nanoparticles, whereas modification with –$CH_3$ and –$PPh_2$ groups is not conducive to the adsorption of nanoparticles [32].

Based on the above considerations, firstly, three organosilicon derivatives containing -SH, -$NH_2$, and epoxy groups were introduced for modification in the preparation of lavender oil fragrance microcapsules based on EC/$SiO_2$ composite shell materials. Subsequently, the effects of different shell modifications on the thermal properties of microcapsules, such as maximum weight loss, temperature, and fragrance retention, were investigated and compared by a scanning electron microscope (SEM), thermo-gravimetric analyzer (TGA), and ultraviolet-visible spectrophotometer (UV). Finally, the differences in heat-resistance properties and aroma release of cotton fibers loaded with the above microcapsules were investigated by gas chromatography –mass spectrometry (GCMS) and E-nose.

## 2. Materials and Methods

### 2.1. Materials and Instruments

Tetraethoxysilane (TEOS), ethyl cellulose (EC), ethyl acetate, sodium dodecyl benzene sulfonate (SDBS), (3-aminopropyl)triethoxysilane (APTES), triethoxy(3-glycidyloxypropyl)silane (GPTES), (3-mercaptopropyl)triethoxysilane (MPTES), and a polypropylene bag (PP bag) were used. All reagents were analytically pure, and purchased from Shanghai Titan Technology Co., Ltd. (Shanghai, China). Lavender fragrance oil (LFO) was prepared by our laboratory. Cotton fabrics were purchased from Shaoxing Manheng Textile Co., Ltd. (Shaoxing, China).

Dynamic light scattering (DLS, Nano Brook Omni, Brookhaven Instruments, New York, NY, USA); a Zeiss Gemini 300 scanning electron microscope (Zeiss, Oberkochen, Germany); a TGA5500 thermal analyzer (TA Instruments, New Castle, DE, USA); an energy-dispersive spectroscope (EDS, Oxford Xplore, Oxford, UK); a Nicolet iN10 Fourier transform micro infrared imaging spectrometer (FTIR, Thermo Fisher Scientific, Waltham, MA, USA); a pyrolysis instrument (PY, EGA/PY-3030D, Frontier LAB, Fukushima, Japan); thermal stability analysis (TGA, TGA5000, TA Instrument, New Castle, DE, USA); and an ultraviolet–visible spectrophotometer (UV, U-3900, Hitachi, Tokyo, Japan) were used.

### 2.2. Preparation of Ethyl Cellulose/Silica Hybrid Microcapsules

During the emulsion solvent diffusion, microcapsules were obtained based on the reported method [33]. Firstly, EC (0.15 g), appropriate TEOS, LFO, and a silane coupling agent (GPTES, APTES, or MPTES) were added into 3 mL of ethyl acetate and stirred to dissolve them evenly. The above solution was slowly added dropwise to the package containing 0.2 g SDBS and ethyl acetate aqueous solution by peristaltic pump, stirred con-

tinuously to form a uniform solution, and sheared at a high speed with a high-speed shear at 10,000 rpm for 15 min. Then, the resulting emulsion was poured into 60 mL of deionized water and stirred for 30 min to make the cellulose/silica material with encapsulated LFO (Figure 1), centrifuged for 3 min at 8000 rpm, then washed with deionized water after freeze drying at −58 °C for 24 h. For comparison, a blank microcapsule without LFO was prepared via the same above method.

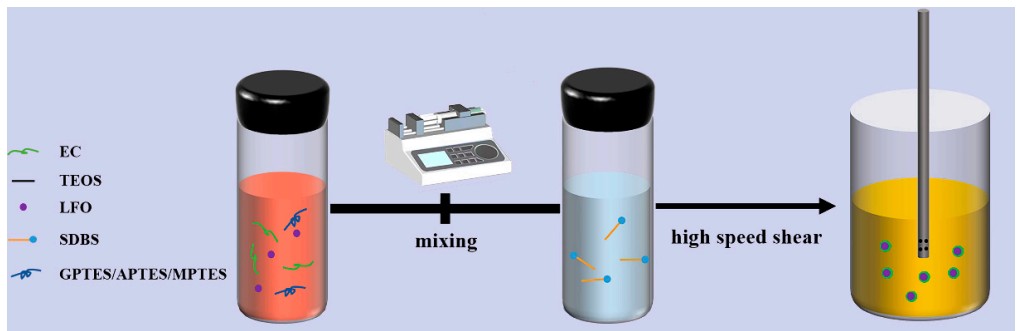

**Figure 1.** The schematic for ethyl cellulose/silica hybrid microcapsule.

### 2.3. Method of Loading Ethyl Cellulose/Silica Fragrance Microcapsules on Cotton Fabrics

The cotton fabric was cut into small pieces (5 × 5 cm$^2$) with scissors. The cotton fabric was dipped into the modified EC/SiO$_2$ fragrance microencapsulated emulsion (solid content: 10 wt%) for 1 h. After that, the dipped cotton fabric was placed in an oven at 120 °C for 30 min by a hot air oven and taken out into a PP bag after the cotton fabric was completely dried.

### 2.4. Test and Characterization

#### 2.4.1. Measurement of Particle Size and Morphology

The particle-size distribution of the microcapsules was determined by dynamic light scattering (DLS).

The microcapsule was bonded to the conductive resin on the metal post. Before observation, the samples were plated with a gold sputter coating machine in a high vacuum evaporator. After pretreatment, the morphology of microcapsules was observed by a scanning electron microscope (SEM) with an accelerated voltage of 15 kV, and the surface elements were analyzed by an energy-dispersive spectrometer (EDS) on a loaded scanning electron microscope.

#### 2.4.2. Thermal Performance Measurement

Thermal stability analysis (TGA, TGA5000, USA) was used to determine the thermal stability of wall materials and three kinds of EC/SiO$_2$ fragrance microcapsules modified by three kinds of silane coupling agents with different EC/TEOS ratios. The experiment was carried out in nitrogen atmosphere at a heating rate of 10 °C/min from 30 °C to 600 °C. The heating curve was recorded, and the thermal degradation behavior was analyzed according to the beginning of weightlessness and the peak value of relative temperature.

#### 2.4.3. FTIR and XPS Characterization of EC/SiO$_2$ Shell Materials

With an appropriate amount of shell materials and microcapsules, the transmittance curves of the materials and microcapsules in the wavelength range of 4000–600 cm$^{-1}$ were obtained by the KBr compression method in Fourier transform infrared imaging spectrometer (FTIR) and a room temperature of 25 °C.

#### 2.4.4. Determination of Slow Release Properties of Fragrance Microcapsules

Standard curve plotting of LFO: the LFO was diluted with anhydrous ethanol and the standard solution with a different concentration gradient was prepared. The solution

was placed into a colorimetric dish and scanned by ultraviolet full spectrum (UV) with a wavelength range of 200–400 nm. The maximum absorption wavelength was detected, the standard curve of absorbance versus concentration was obtained, and the linear correlation coefficient was calculated.

The microencapsulated sample of 15 mg was added to anhydrous ethanol (5 mL). After it was completely dissolved, the absorbance of the sample was determined. The essence microcapsules were placed in the oven at 120 °C and sampled every 0, 1, 2, 3, 4, 5, and 6 h. The curve of fragrance content with time was obtained, and the release amount of fragrance was calculated according to the standard curve of LFO.

2.4.5. Determination of Fragrance Microcapsules by Pyrolysis-Gas Chromatography—Mass Spectrometry (PY-GCMS)

Appropriate samples were added to the middle position of the pyrolysis quartz tube, and then the quartz tube was placed into the pyrolysis instrument (PY, EGA/PY-3030D, Japan). The pyrolysis was carried out in an air atmosphere at a set temperature of 120 °C., and the pyrolysis product (GC-MS, QP2010, Japan) was collected by a pyrolysis bottle for 20 s. After that, the injection port of gas chromatography was connected, and the pyrolysis products were separated by gas chromatography and identified by mass spectrometry. Gas chromatography—mass spectrometry conditions: the flow rate of the carrier gas (hydrogen) was 1.0 mL/min, the injection temperature was 230 °C, the temperature program was 50 °C (keep 5 min) to 100 °C (keep 1 min) and risen to 280 °C (keep 10 min) at 15 °C/min, and the split ratio was 1: 10; MS condition: the interface temperature was 230 °C, the EI source temperature was 250 °C, the ionization energy was 70 eV, and the mass scanning range was 3500 u.

2.4.6. Detection of Changes in Characteristic Aroma Substances in Scented Cotton Fabrics

The aromatic cotton fabric of $1.00 \pm 0.01$ g was accurately weighed, shredded, and put into a headspace bottle, and 50 μL of ethyl decanoate with a concentration of 400 mg/L was added as the internal standard. It was quantitatively analyzed by gas chromatography–mass spectrometry (GCMS).

The changes of embedded LFO and characteristic aroma substances during the release process were determined by GCMS. The conditions of the GCMS instrument were as follows: the gas chromatographic column was DB-5 capillary column (60 m × 0.25 mm × 0.25 μm, Agilent Technologies, Palo Alto, CA, USA), the inlet temperature was 250 °C, the ion source temperature was 230 °C, the solvent delay time was 3 min, the carrier gas was helium, and there was no shunt. The heating procedure was as follows: the initial temperature was 50 °C; after 5 min, the temperature rises to 140 °C at the rate of 5 °C/min for 5 min, and then rises to 230 °C at the rate of 3 °C/min for 10 min. The electron energy of mass spectrometry is 69.99 eV.

2.4.7. E-nose Testing of Fragrance-Laden Cotton Fabrics

The aromas of the fabrics were determined using the E-nose (Heracles II, Alpha Mos, Toulouse, France). E-nose contain two chromatographic MXT-5/MXT-1701 (length: 10 m; column diameter: 180 μm). Alpha soft V12.44 was used for data processing. The injection volume was set at 800 μL, the injection rate at 125 μL/s, and the inlet temperature at 200 °C.

## 3. Results and Discussion

### 3.1. Effect of EC/TEOS Addition Ratio on Heat Resistance of Wall Material

Figure 2a shows TG and DTG of the GCSM (GCSM: EC/SiO$_2$ composites modified by GPTES), and the maximum weight loss rate (T$_{max}$) of the composites obtained at different EC/TEOS addition ratios is shown in Table 1. With the increase in TEOS, the thermal degradation process can be divided into two processes. Approximately 5% of the weight loss rate occurred between 0 °C to 250 °C, which is due to the evaporation of residual water in the product at high temperatures, and 75% of the weight loss between 250 °C to 350 °C

is due to the decomposition of EC. The weight loss between 350 °C to 600 °C is mainly due to the decomposition of $SiO_2$ in the wall material [33].

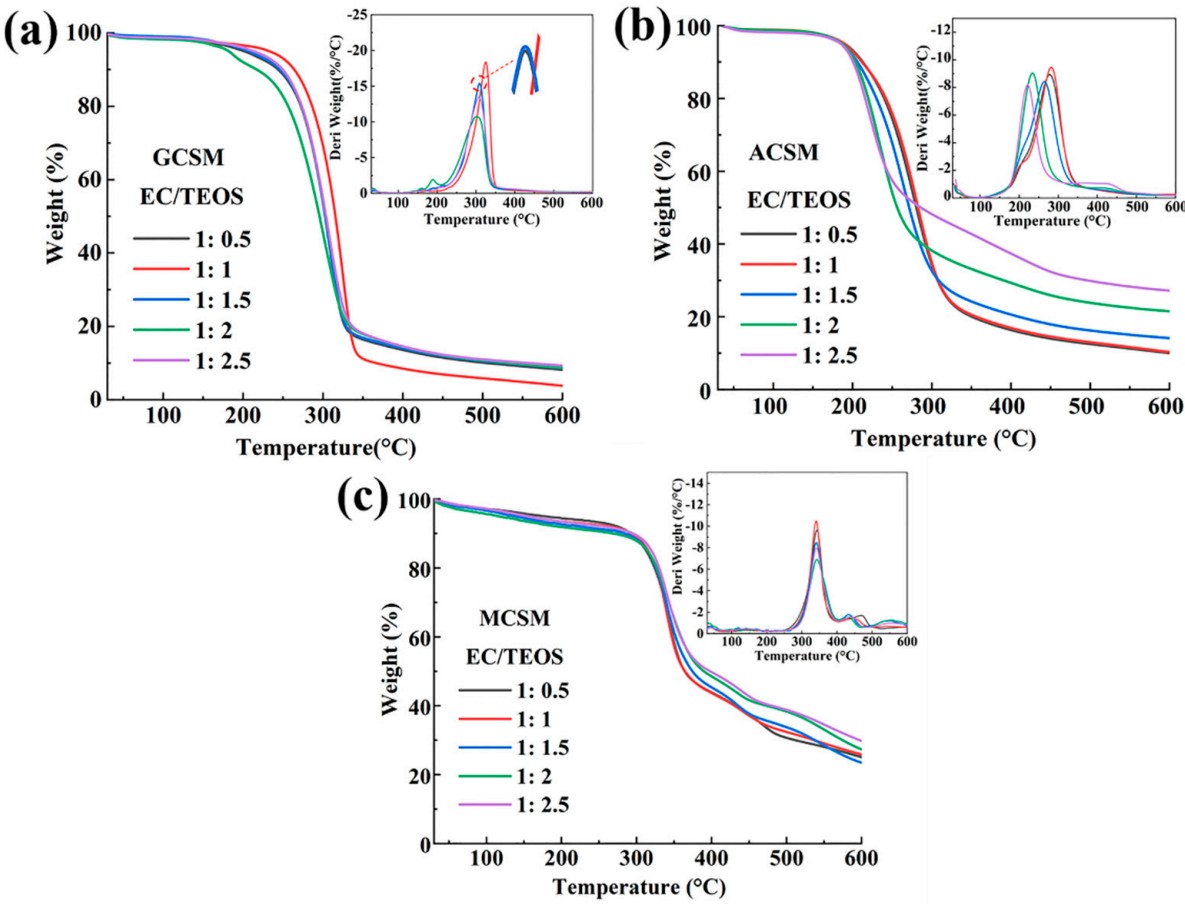

**Figure 2.** (**a**) TG curves (from 30 °C to 600 °C) of GCSM; (**b**) TG curves (from 30 °C to 600 °C) of ACSM; (**c**) TG curves (from 30 °C to 600 °C) of MCSM.

**Table 1.** $T_{max}$ of the wall material at different EC/TEOS addition ratios.

| EC/TEOS | $T_{max}$ (°C) GPTES | APTES | MPTES |
|---|---|---|---|
| 1:0.5 | 309.03 | 215.70 | 342.20 |
| 1:1 | 325.40 | 213.17 | 340.76 |
| 1:1.5 | 309.00 | 217.51 | 340.60 |
| 1:2 | 308.00 | 212.35 | 342.80 |
| 1:2.5 | 302.00 | 213.07 | 340.47 |

Figure 2b shows the TG and DTG of ACSM (ACSM: EC/$SiO_2$ composites modified by APTES), and the maximum weight loss rate $T_{max}$ is shown in Table 1. A small amount of weight loss at 0–200 °C is also due to the evaporation of residual water in the product, and approximately 80% of the weight loss between 200 °C to 300 °C is due to EC thermal decomposition, and after 300 °C it is the same as above.

Figure 2c shows the TG and DTG of MCSM (MCSM: EC/$SiO_2$ composites modified by MPTES). $T_{max}$ is relatively large at 340.3 °C for an EC/$SiO_2$ of 1:2. The weight loss at 0–300 °C is the same as residual water, and the decomposition of EC is at 300–370 °C. The cause of weight loss after 370 °C is the same as above.

Based on the TGA and DTG results, higher $T_{max}$ and lower maximum weight loss rate leads to a better heat resistance of the shell material. As a result, it is concluded that the best proportion of EC/$SiO_2$ composites is 1:1, 1:1.5, and 1:2 when modified with the silane

coupling agents GPTES, APTES, and MPTES, respectively. Therefore, microcapsules are prepared at these shell ratios.

### 3.2. FTIR Spectra of Modified EC/SiO$_2$ Shell Materials

The results of Fourier transform infrared spectroscopy (FTIR) of the modified EC/SiO$_2$ materials are shown in Figure 3. It can be observed from the figure that after silanization, the stretching vibration peak at 2973 cm$^{-1}$ corresponds to -CH$_2$ and 1450 cm$^{-1}$ corresponds to the C-H shear vibration peak. A clear absorption peak at 1103 cm$^{-1}$ is produced by the Si-O-Si stretching vibration in the silica nanostructure, and at 764 cm$^{-1}$ is due to the symmetric stretching vibration of Si-O. The figure also shows the characteristic peaks of epoxy groups (918 cm$^{-1}$), -NH$_2$ (1630 cm$^{-1}$), and -SH (2553 cm$^{-1}$) for three different silane coupling agents, as shown by the red circle mark in the figure, indicating the successful polycondensation reaction of TEOS with GPTES, APTES, and MPTES, respectively [27].

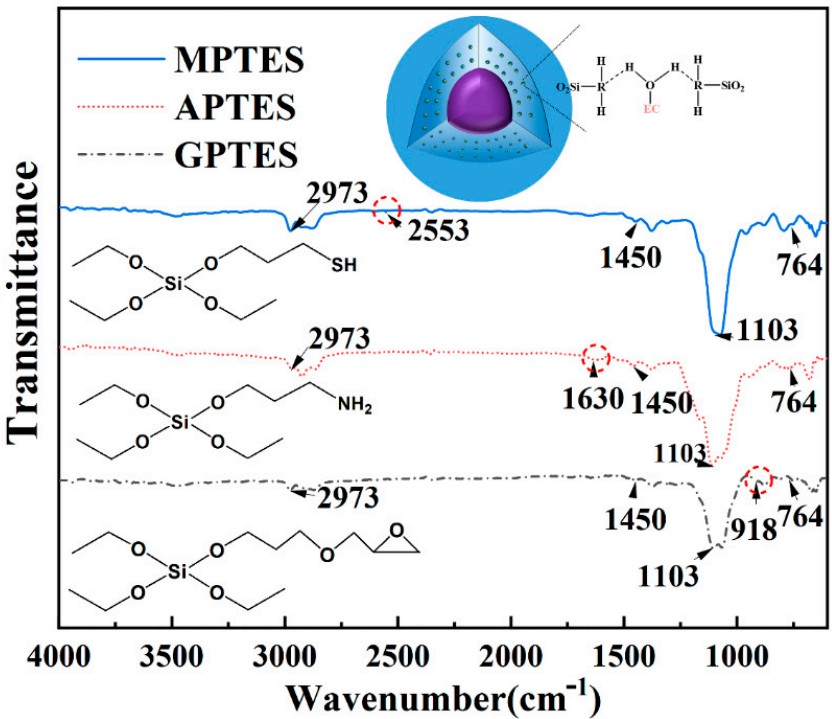

**Figure 3.** FTIR diagram of the three modified EC/SiO$_2$ shell materials.

### 3.3. Morphological Analysis of Modified EC/SiO$_2$ Fragrance Microcapsules

Figure 4a shows the particle-size distribution of EC/SiO$_2$ fragrance microcapsules modified with GPTES, APTES, and MPTES silane coupling agents at EC/TEOS ratios of 1:1, 1:1.5, and 1:2. In Figure 4a, GPTES-modified EC/SiO$_2$ fragrance microcapsules were found to be distributed between 474 nm to 698 nm with an average particle size of 534 nm; APTES-modified EC/SiO$_2$ fragrance microcapsules were distributed between 474 nm to 1507 nm with an average particle size of 836 nm; and MPTES-modified EC/SiO$_2$ fragrance microcapsules were distributed between 384 nm to 572 nm with an average particle size of 460 nm. In addition, it can be clearly seen in Figure 4b–d that the microcapsules are spherical and the surface of the microcapsules is slightly rough and dense under the polycondensation of TEOS. Some larger microcapsules appear in the SEM images, probably attributed to inadequate homogenization during the preparation process. In Figure 4b, there is a ruptured microcapsule, which can further confirm the core-shell structure of the microcapsule. Further measurement by EDS spectrum showed that the shell of the microcapsule was composed of C, O, N, and Si.

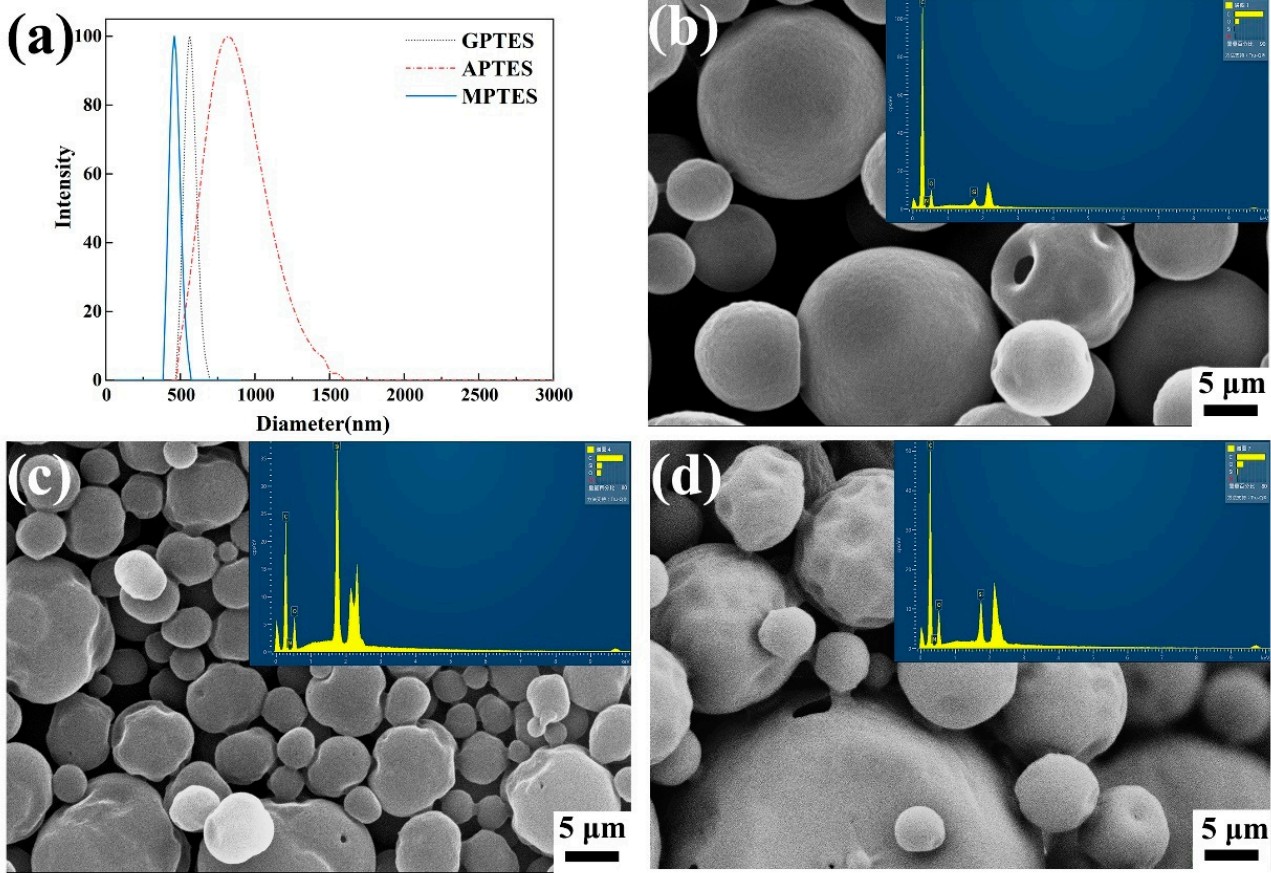

**Figure 4.** (**a**) Particle-size distribution of different modified essence microcapsules; (**b**) SEM image of GPTES-modified EC/SiO₂ fragrance microcapsules; (**c**) SEM image of APTES-modified EC/SiO₂ fragrance microcapsules; (**d**) SEM image of MPTES-modified EC/SiO₂ fragrance microcapsules.

*3.4. Thermal Performance Analysis of Fragrance Microcapsules*

Figure 5 shows the thermogravimetric (TG) analysis of blank samples of ethyl cellulose fragrance microcapsules without the addition of modified silica and modified EC/SiO₂ fragrance microcapsules. The fragrance in the modified fragrance microcapsules decreased with the increase in temperature. From the TG curve, it can be seen that there was approximately 2% weight loss in the microcapsule modified by APTES. According to the TG curve of LFO, this stage is caused by the moisture and essence on the surface of the microcapsules. The second stage of weight loss was from 100 °C to 200 °C, and its mass loss was about 10%, which can be known as 10% loading of fragrance. Although the shell material started to lose weight only from 290 °C, the weight change from 200 °C to 290 °C was not obvious, indicating that the fragrance was volatilized until 200 °C. The TG curve of GPTES-modified fragrance microcapsules showed that there was approximately a 2% weight loss of water and fragrance on the surface of the microcapsules from 0 °C to 100 °C. The mass loss ratio of approximately 13% between 100 °C to 220 °C can be known to be approximately 13% loading of the microcapsules. Compared with others, the mass loss ratio of MPTES-modified fragrance microcapsules was 19% in the second weight loss stage, and the fragrance was not completely volatilized in the microcapsules until 252 °C. However, the blank sample was volatilized by 170 °C after the water and fragrance evaporated from the microcapsule surface before 100 °C, and the encapsulated LFO was already volatilized by 170 °C. According to the TG diagram of the fragrance microcapsules, the order of the thermal protection ability of the microcapsules for the fragrance after the modification of EC/SiO₂ by three silane coupling agents should be MPTES, GPTES, and APTES.

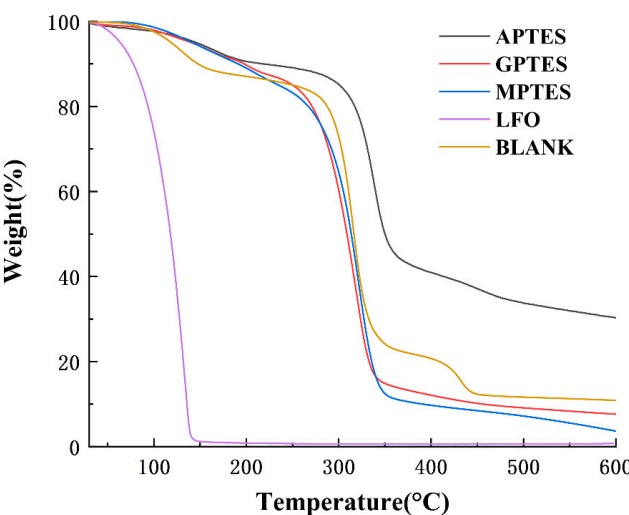

**Figure 5.** TGA curves of modified EC/SiO$_2$ fragrance microcapsules, LFO and EC fragrance microcapsules (blank).

The standard curve of LFO are shown in Figure 6a. The absorbance of the sample was measured at the maximum absorption wavelength, and the amount of LFO can be calculated from the standard curve. The release behavior of fragrance microcapsules at a high temperature is shown in Figure 6b, comparing the effect of different silane coupling agents with ethyl cellulose fragrance microcapsules on the fragrance. The amount of fragrance in the microcapsules gradually decreased with time. In Figure 6b, the release rate of fragrance microcapsules without added silica reached 33% after 2 h treatment at 120 °C and 38.9% for 6 h. The three different silane coupling agents, GPTES, APTES, and MPTES, modified the shell material, and the fragrance release rates were 25.2%, 35.1%, and 16.7% after treatment of the microcapsules at 120 °C for 6 h. This demonstrated that the addition of modified SiO$_2$ to EC enhanced the slow release of fragrance in microcapsules at high temperatures. Combined with the results of the FTIR analysis, the reason for the difference in the fragrance release results may be due to the introduction of -NH$_2$ in APTES. Although it can improve the heat-resistant type of raw materials to some extent, the epoxy group and -SH in GPTES and MPTES combined with EC to form a wall material that protects the fragrance significantly more than -NH$_2$. It indicates that EC and SiO$_2$, after GPTES and MPTES modified and combined, the formed fragrance microcapsules have a significant slow-release effect and have a better protection effect on the fragrance.

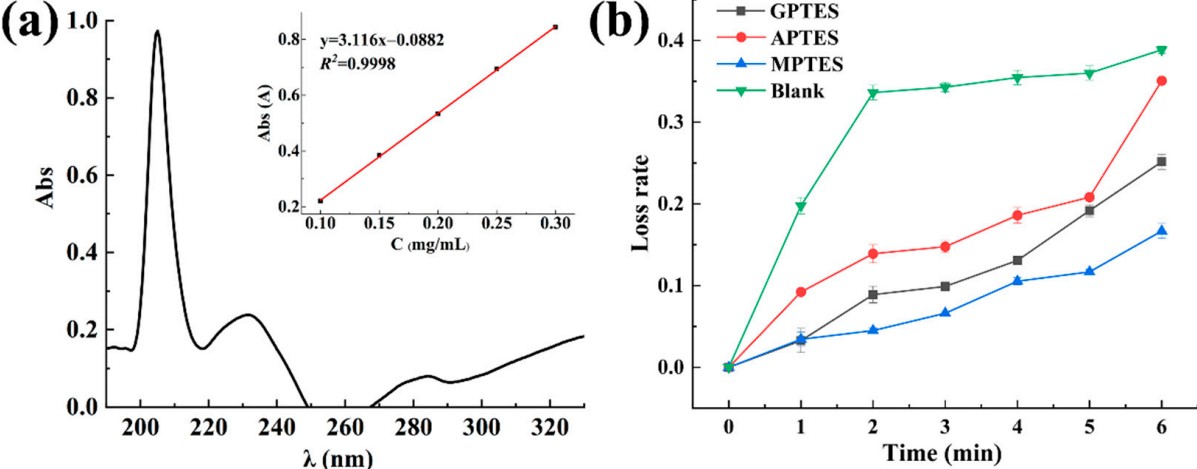

**Figure 6.** (**a**) Standard curve for LFO; (**b**) UV curves of fragrance content of modified EC/SiO$_2$ fragrance microcapsules and EC fragrance microcapsules treated at 120 °C.

*3.5. Py-GCMS Analysis of EC/SiO$_2$ Fragrance Microcapsules*

To determine whether the heating treatment of microcapsules in the rinsing, disinfection, and printing processes after fragrance addition to fabrics would have an effect on the aroma, the thermal cleavage behavior of the microcapsules was investigated. The results of the thermal cleavage products analysis of three different silane coupling agent-modified fragrance microcapsules and LFO are shown in Table 2. The thermal cleavage products at 120 °C were mainly linalyl acetate, linalool, methyl dihydrojasmonate, laevo-camphor, and lavandulyl acetate. The thermal cleavage products of GPTES-modified EC/SiO$_2$ fragrance microcapsules at 120 °C were mainly methyl dihydrojasmonate, methyl 4,6-di-O-methyl-α-D-mannopyranoside, linalylacetate, and geraniol acetate (Table 3), which indicated that the aroma of the fragrance microcapsules at 120 °C did not differ much from that of LFO. The main thermal cleavage products of APTES-modified EC/SiO$_2$ fragrance microcapsules are shown in Table 3, and the main cleavage products do not differ much from those of LFO. However, due to the presence of -NH$_2$ in APTES, Ala-Ala-Ala was produced to affect the aroma of the fragrance microcapsules. The main thermal cleavage products of MPTES-modified EC/SiO$_2$ fragrance microcapsules are shown in Table 3, and their main aroma components are still mainly methyl dihydrojasmonate, thymol, linalyl acetate, and linalool, which is not much different from the main products of the thermal cleavage of LFO.

**Table 2.** Py-GCMS results of LFO at 120 °C.

| 300 °C | |
|---|---|
| **Compound Name** | **Area%** |
| Linalyl acetate | 40.61 |
| Linalool | 24.93 |
| Eucalyptol | 11.42 |
| Methyl dihydrojasmonate | 4.27 |
| Laevo-camphor | 4.01 |
| Terpinen-4-ol | 1.26 |
| 5-Isopropyl-2-methylphenol | 1.44 |
| Lavandulyl acetate | 2.75 |
| Caryophyllene | 1.35 |
| Myrcene | 1.17 |
| Neryl acetate | 0.96 |
| Trans-linalool oxide (furanoid) | 0.66 |
| Trans-β-farnesene | 0.56 |

**Table 3.** Py-GCMS results of different modified EC/SiO$_2$ fragrance microcapsules.

| GPTES | | APTES | | MPTES | |
|---|---|---|---|---|---|
| **Compound Name** | **Area%** | **Compound Name** | **Area%** | **Compound Name** | **Area%** |
| Methyl dihydrojasmonate | 17.33 | Methyl dihydrojasmonate | 15.32 | Methyl dihydrojasmonate | 22.37 |
| Thymol | 12.59 | Ala-ala-ala | 0.47 | Methyl 3-O-acetyl-2,4,6-tri-O-Benzyl-β-D-mannopyranoside | 22.87 |
| Methyl 4,6-di-O-methyl-α-D-Mannopyranoside | 13.8 | Thymol | 11.71 | Thymol | 9.14 |
| Thiolutin | 2.81 | Linalyl acetate | 5.27 | Linalyl acetate | 6.53 |
| Linalyl acetate | 2.29 | Linalool | 1.42 | Geraniol acetate | 1.93 |
| 2,2-Dimethylsuc cinic acid | 2.08 | Caryophyllene | 1.26 | Myrcene | 1.7 |
| Ethanol | 1.66 | Neryl acetate | 1.08 | Neryl acetate | 1.6 |
| β-Caryophyllene | 1.66 | Phosphinic acid, diphenyl | 0.73 | Linalool | 1.56 |
| Geraniol acetate | 1.18 | Trans-β-farnesene | 12.7 | (E)-β-Famesene | 1.17 |
| Myrcene | 1.05 | (-)-Isopulegol | 0.33 | Ethanol | 0.68 |
| Linalool | 0.52 | | | β-Ocimene | 0.57 |
| (Z)-beta-farnesene | 0.34 | | | | |

In summary, the main aroma substances in the thermal cleavage products of the three fragrance microcapsules at 120 °C were slightly different compared with LFO, which was due to the shell material of the microcapsules hindering the volatilization of the aroma components, causing a slight effect on the thermal cleavage products. In addition, the -NH$_2$ in APTES causes an unpleasant odor to the fragrance. After comparing the thermal

cleavage products of the three microcapsules, the MPTES-modified EC/SiO$_2$ fragrance microcapsules had less effect on the aroma at 120 °C.

### 3.6. Aroma Analysis of Microcapsules before and after Heat Treatment

Figure 7 compares the E-nose radar images of three kinds of modified EC/SiO$_2$ fragrance microcapsules and EC shell-material fragrance microcapsules before and after treatment at 120 °C. Figure 7a shows the aroma curves of four kinds of microcapsules before heat treatment, and Figure 7b shows the radar map of the aroma after treatment at 120 °C for 6 h. In Figure 7a, there are several peaks with high content and complex components, and the peaks of the four kinds of microcapsules almost coincide, which indicates that the fragrance microcapsules of modified microcapsules and ethyl cellulose shell materials are almost the same before heat treatment. In Figure 7b, the peaks of many components disappeared or decreased, indicating the volatilization or loss of small molecular aroma components. The peaks of many components disappeared or decreased after ECMF was treated at 120 °C for 6 h. This indicated that the EC shell material could not protect the small molecule aroma component in the fragrance better, which led to the volatilization or loss of aroma comments. Observe the heat-treatment results of GCSMF, ACSMF, and MCSMF: the characteristic aroma losses are less than ECMF. The above results show that the aroma retention value of silane coupling agent-modified composite shell-material fragrance microcapsules after scenting the cotton fabric is better than that of EC shell-material fragrance microcapsules.

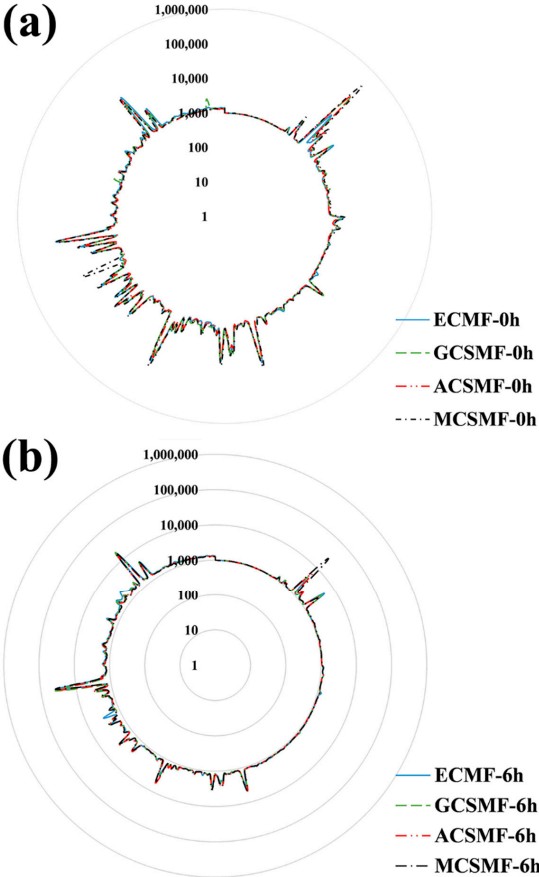

**Figure 7.** (**a**) Comparison of E-nose between EC microcapsules scented cotton fabric (ECMF), EC/SiO$_2$ hybrid microcapsules by GPTES-modified scented cotton fabric (GCSMF), EC/SiO$_2$ hybrid microcapsules by APTES-modified scented cotton fabric (ACSMF), and EC/SiO$_2$ hybrid microcapsules by MPTES-modified scented cotton fabric (MCSMF); (**b**) comparison of E-nose between ECMF, GCSMF, ACSMF, and MCSMF after treatment at 120 °C.

Five typical aroma substances of LFO were selected in the experiment. As shown in Figure 8, when the four kinds of fragrance microencapsulated cotton fabrics were treated at 120 °C for 6 h, the release of characteristic aroma substances in ECMF was the highest, and the addition and modification of SiO$_2$ had an obvious protective effect on the characteristic aroma substances of LFO. This conclusion corresponds to the UV analysis of fragrance microcapsules. Methyl 2-(3-oxo-2-pentylcyclopentyl) acetate is the substance with the highest content of aroma components in LFO, and the protective ability of this aroma substance is GPTES, MPTES, and APTES in turn. However, generally speaking, the three silane coupling agents have their own advantages in protecting LFO.

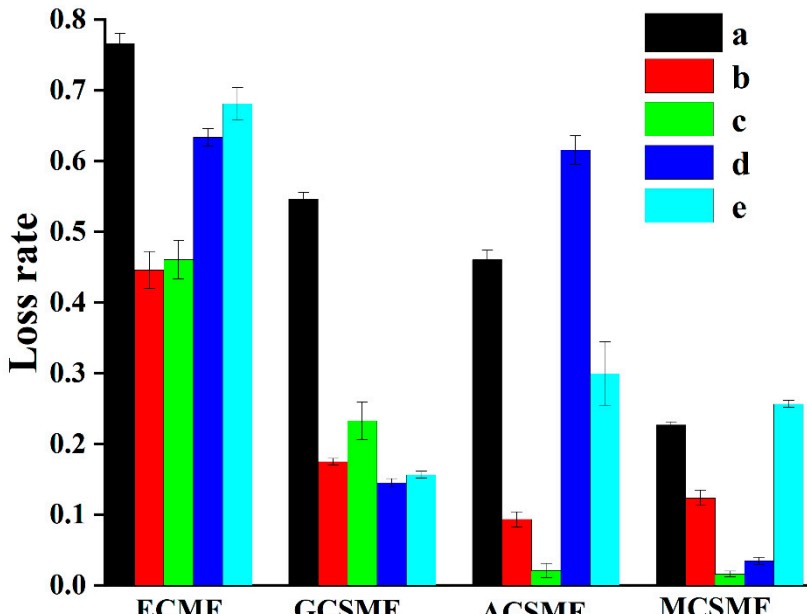

**Figure 8.** GCMS analysis of characteristic aroma components of scented cotton fabric after heat treatment with ECMF, GCSMF, ACSMF, and MCSMF: (a) β-caryophyllene; (b) limonene; (c) thymol; (d) acetic acid geranyl ester; (e) methyl 2-(3-oxo-2-pentylcyclopentyl) acetate.

All in all, from the E-nose and GC-MS analysis, we can know that the addition of modified silica has an aroma protection effect on scented cotton fabric, and the number of volatile aroma components in the microcapsules modified by silane coupling agents is higher than that of ethyl cellulose wall materials, which further proves that the modified composite shell-material microcapsules can effectively protect the aroma substances of LFO and slow down the volatilization of LFO at high temperatures. Compared with the volatilization of aroma components of each kind of microcapsule, the comprehensive retention value of volatile aroma components of MPTES-modified microcapsules was the best.

## 4. Conclusions

In this study, EC/SiO$_2$ fragrance microcapsules modified by GPTES, APTES, and MPTES were prepared by a one-step emulsion solvent diffusion method, and the difference of the heat resistance of modified composite shell materials was investigated. TGA showed that the thermal decomposition temperatures of the three modified wall materials were MPTES > GPTES > APTES. After the fragrance was embedded, the protective ability of the three modified shell materials to fragrance at high temperatures was MPTES > GPTES > APTES. The results show that the existence of -SH combined with EC has a strong ability to protect the fragrance. The pyrolysis results showed that the modified microcapsules had an effect on the aroma of the essence at 900 °C, among which APTES had a greater effect. At 300 °C, except for APTES, the pyrolysis products of the other two

fragrance microcapsules were almost the same as those of LFO. In the future, further research can be carried out on the interaction between fragrance and modified shell materials and the controlled release of fragrance.

**Author Contributions:** Funding acquisition, Z.X.; writing—review and editing, B.Z.; methodology, X.K.; project administration, Y.N.; writing—review and editing, X.L.; investigation, L.H., H.C. and W.Z. All authors have read and agreed to the published version of the manuscript.

**Funding:** This research was funded by the Program of Shanghai Academic Research Leader (21XD1423800), Capacity building project of the local universities' Science and Technology Commission of Shanghai Municipality (21010503900), and the key project of China Tobacco Yunnan Industrial Co. LTD (2021XY04).

**Institutional Review Board Statement:** Not applicable.

**Informed Consent Statement:** Not applicable.

**Data Availability Statement:** The data that support the findings of this study are available from the corresponding author, L.X., upon reasonable request.

**Conflicts of Interest:** The authors declare no conflict of interest.

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
