# Peer review of "High-Temperature Aroma Mitigation and Fragrance Analysis of Ethyl Cellulose/Silica Hybrid Microcapsules for Scented Fabrics"

_coatings, doi:10.3390/coatings12050711_

Round 1
Reviewer 1 Report
The reviewer’s comments are as below:
- Page 3 of 24, Section 1. Introduction: The author selected -SH, -NH2, and epoxy groups for the modification of microcapsule shell materials. It is recommended to highlight the rational or a scientific reason (with references if there are any) for selecting these functional groups. Currently, the manuscript is lacking the basis for investigating only these specific groups. Although it is not necessary but it would be good to include a table showing other functional groups based on literature in comparison to these groups and their impacts.
- Section 2.3, page 4 of 24: “The cotton fabric was dipped into the modified EC/SiO2 fragrance microencapsulated emulsion for 1 h”, What was the concentration of this emulsion and how the consistency from one sample to another was ensured? Further, these cotton fabrics were dried at 120C. How long this drying took place, and was it a vacuum oven or a hot air oven? In either case, how it may impact the final results?
- Section 2.4.2, page 5 of 24: The TGA experiment was conducted under Nitrogen. Any specific reason, why not under oxygen/air? In real applications, it is more of air conditions rather Nitrogen. Later in section 2.4.5, the pyrolysis was conducted in air atmosphere, why?
- Section 3.1, page 9 of 24: “As a result, it is concluded that the best proportion of EC/SiO2 composites is 1:1, 1:1.5 and 1:2 when modified with silane coupling agent GPTES, APTES and MPTES, respectively. Therefore, microcapsules are prepared at these shell ratios.”, These are quite interesting results, however is there any specific explanation for different optimum ratios for different functional groups?
- Revise the keywords. For examples “modified” itself is not a proper keyword. Please review the manuscript with respect to any typo or grammatical errors. There are some instances such as –
- “composite materials as shell materials are one way to further improve the thermal”
- “An uniform solution”
- “GTPES-Modified”
- “MPTES, GTPES, and ATPES”, etc.
- It would be good to include a future research study recommendation in the “Conclusion” section based on the current findings.
Author Response
Thank you very much for your help in processing the review of our manuscript. On the basis of the enlightening questions and helpful advises, we have now completed the revision of our manuscript.
Please see the attachment.

Reviewer 2 Report
Authors have presented an article entitled “High-temperature aroma mitigation and fragrance analysis of ethyl cellulose/silica hybrid microcapsules for scented fabrics” Though the manuscript is well written and organized but there is scope for further improving the quality of the draft before considering for publication.
Detail minor comments are listed below:
- As the author mentioned that “Cotton fabrics with different filler decoration having multiple applications”. Author should send the performance of these articles to make the contrast: Fibers and Polymers 20 (6), 1161-1171 (2019); Composites Science and Technology 181, 107682 (2019).
- Authors are requested to combine Figure (2-4) and make a single Figure, as it will be easy to see the comparison.
- In Figure 6, the scale bars are not visible properly. Make it bold.
- In SEM micrographs, there are few small and big spherical architecture recorded. Big architecture formed due to agglomeration or anything else. Please mention.
- Authors have nicely presented the experimental part and results and discussion. They should also check the typos and grammatical mistakes in the revised version.
Author Response
Thank you very much for your help in processing the review of our manuscript. On the basis of the enlightening questions and helpful advises, we have now completed the revision of our manuscript.
Please see the attachment.

This manuscript is a resubmission of an earlier submission. The following is a list of the peer review reports and author responses from that submission.